

# Immune response of human cultured cells towards macrocyclic $Fe_2PO$ and $Fe_2PC$ bioactive cyclophane complexes

Alex J. Salazar-Medina[1], Enrique F. Velazquez-Contreras[2,3], Rocio Sugich-Miranda[3], Hisila Santacruz[2], Rosa E. Navarro[2], Fernando Rocha-Alonzo[3], Maria A. Islas-Osuna[2,4], Patricia L. Chen[5], Sarah G.B. Christian[5], Amelia A. Romoser[6], Scott V. Dindot[6], Christie M. Sayes[7], Rogerio R. Sotelo-Mundo[8] and Michael F. Criscitiello[5]

[1] Cátedra CONACYT-Departamento de Investigación en Polímeros y Materiales, Universidad de Sonora, Hermosillo, Sonora, Mexico
[2] Departamento de Investigación en Polímeros y Materiales, Universidad de Sonora, Hermosillo, Sonora, Mexico
[3] Departamento de Ciencias Químico Biológicas, Universidad de Sonora, Hermosillo, Sonora, Mexico
[4] Centro de Investigación en Alimentación y Desarrollo, A.C., Hermosillo, Sonora, Mexico
[5] Comparative Immunogenetics Laboratory, Department of Veterinary Pathobiology, Texas A&M University, College Station, TX, USA
[6] Department of Veterinary Pathobiology, Texas A&M University, College Station, TX, USA
[7] Nanotoxicology and Nanopharmacology, RTI International, Research Triangle, MC, USA
[8] Biomolecular Structure Laboratory, Centro de Investigación en Alimentación y Desarrollo, A.C., Hermosillo, Sonora, Mexico

Corresponding authors
Rogerio R. Sotelo-Mundo, rrs@ciad.mx
Michael F. Criscitiello, mcriscitiello@cvm.tamu.edu

## ABSTRACT

Synthetic molecules that mimic the function of natural enzymes or molecules have untapped potential for use in the next generation of drugs. Cyclic compounds that contain aromatic rings are macrocyclic cyclophanes, and when they coordinate iron ions are of particular interest due to their antioxidant and biomimetic properties. However, little is known about the molecular responses at the cellular level. This study aims to evaluate the changes in immune gene expression in human cells exposed to the cyclophanes $Fe_2PO$ and $Fe_2PC$. Confluent human embryonic kidney cells were exposed to either the cyclophane $Fe_2PO$ or $Fe_2PC$ before extraction of RNA. The expression of a panel of innate and adaptive immune genes was analyzed by quantitative real-time PCR. Evidence was found for an inflammatory response elicited by the cyclophane exposures. After 8 h of exposure, the cells increased the relative expression of inflammatory mediators such as interleukin 1; IRAK, which transduces signals between interleukin 1 receptors and the NFκB pathway; and the LPS pattern recognition receptor CD14. After 24 h of exposure, regulatory genes begin to counter the inflammation, as some genes involved in oxidative stress, apoptosis and non-inflammatory immune responses come into play. Both $Fe_2PO$ and $Fe_2PC$ induced similar immunogenetic changes in transcription profiles, but equal molar doses of $Fe_2PC$ resulted in more robust responses. These data suggest that further work in whole animal models may provide more insights into the extent of systemic physiological changes induced by these cyclophanes.

## INTRODUCTION

Transition metal complexes, natural and synthetic, are widely distributed and are notable for the great variety of functions they can perform (*De Souza & De Giovani, 2004*). Among them, the metalloenzymes involved in defense against reactive oxygen species (ROS), are key to survival in the aerobic environments of organic life. Superoxide dismutase, peroxidase and catalase have in their active centers transition metal ions of Cu, Fe, Mn, Ni and Zn, to name but a few (*Rosati & Roelfes, 2010*). The use of biologically-inspired synthetic complexes is a strategy to develop new materials designed to mimic the function of their natural analogs. However, a compound, natural or synthetic, could affect the gene expression of cells, often through immunologic pathways. We used human embryonic kidney (HEK) 293T cells exposed to the metal complexes $Fe_2$-2,9,25,32-tetraoxo-4,7,27,30-tetrakis (carboxymethyl)-1,4,7,10,24,27,30,33-octaaza-17,40-dioxa (10.1.10.1) paracyclophane ($Fe_2PO$) and $Fe_2$-2,9,25,32-tetraoxo-4, 7.27.30-tetrakis (carboxymethyl)-1,4,7,10,24,27,30,33-octaaza (10.1.10.1) paracyclophane ($Fe_2PC$) (*Salazar-Medina et al., 2013*).

Fe$_2$PO and Fe$_2$PC (Fig. 1) are binuclear Fe(III) complexes derived from macrocyclic cyclophane receptors, which by definition include in their structure at least one aromatic ring. The advantages of macrocyclic receptors are that they contain the groups that coordinate the metallic ion combined with the stability of the aromatic rings, increasing the affinity and strength of the complex (*Diederich, 1991*).

The Fe$_2$PO and Fe$_2$PC complexes have been widely studied as antioxidants against synthetic free radicals using in vitro systems, reducing oxidative stress in human peripheral blood mononuclear cells and in murine cell lines (*Salazar-Medina et al., 2013*, *2017*). Also, their ability to mimic the activity of the enzymes superoxide dismutase and peroxidase has been demonstrated by in vitro assays. There is perceived safety in both free receptors and their complexes with Fe(III), while their bioactive properties are attributed exclusively to metal complexes. However, this has not been tested. This work aims to investigate the effects on the inflammatory response of Fe$_2$PO and Fe$_2$PC on human cells, establishing the basis for future in vivo studies.

## MATERIALS AND METHODS

### Cell culture and experimental dosing

Human embryonic kidney (HEK) 293T cells (ATCC, Manassas, VA, USA) were cultured in Dulbecco's Modified Eagle's Medium supplemented with 8% fetal bovine serum (FBS). The media was supplemented with an antibiotic cocktail (penicillin, streptomycin, and amphotericin). HEK cells were grown to 80% confluence in 6-well plates at 37 °C with humidity and 5% $CO_2$, and then treated with an aqueous solution of Fe$_2$PO or Fe$_2$PC, to a final concentration of 120 μM. Sterile milliQ water was used as a negative control. Synthesis and characterization of the Fe$_2$PO and Fe$_2$PC cyclophanes has been

**Figure 1 Graphical representation of binuclear Fe(III) cyclophane complexes.** For Fe$_2$PO, X represents (–O–); for Fe$_2$PC complex, X represents (–CH$_2$–).

previously described (*Salazar-Medina et al., 2013*). Each experiment was performed in duplicate.

## Gene expression analysis

Cells were harvested from each cyclophane complex treatment at time points of 8 and 24 h. The RNeasy Mini Kit (Qiagen-SABiosciences, Frederick, MD, USA), which included a step to remove genomic DNA, and the RT$^2$ First Strand Kit (Qiagen-SABiosciences, Frederick, MD, USA) was used for synthesizing cDNA by reverse transcription. The RT$^2$ Profiler™ PCR Array Human Innate and Adaptive Immune Responses, specific for 84 immune genes (catalog PAHS-052Z; Qiagen-SABiosciences, Frederick, MD, USA) was used to query immune responsive gene expression. This method consists of a quantitative PCR reaction with SYBR-green detection, where the cDNA is loaded into 96-well PCR array plates. Each plate has specific primers for the 84 immune genes, plus housekeeping genes and experimental controls. The protocol and recommendations of the manufacturer were followed thoroughly.

The experiment was performed on a Roche LightCycler 480 (Roche, Indianapolis, IN, USA) thermal cycler and the amplification conditions were: one cycle of 95 °C for 10 min, then 45 cycles of 95 °C for 15 s, and 60 °C for 1 min. Raw data were acquired and analyzed using the Roche LightCycler 480 software, including the calculation of $C_t$ (threshold cycle) values, which are proportional to the abundance of the cDNA of interest (*Romoser et al., 2011*).

## Calculation of gene expression changes

The RT$^2$ Profiler™ PCR Array Human Innate and Adaptive Immune Responses array includes five housekeeping genes: β$_2$-microglobulin, hypoxanthine phosphoribosyltransferase 1 (*HPRT1*), ribosomal protein L13a, glyceraldehyde-3-phosphate dehydrogenase (*GADPH*), and β-actin. Among those, the β$_2$-microglobulin gene (*B2M*) expression showed less variance across the sample conditions. Therefore, *B2M*

was utilized for subsequent normalization. The $2^{-\Delta\Delta Ct}$ method was used to estimate changes in relative gene expression (*Livak & Schmittgen, 2001*), and the fold changes were calculated for the expression level of the genes in the respective control group. For instance, a 2.0-fold change of gene A indicates that the expression of gene A was twice as abundant in the samples treated with the cyclophane compared to the expression in the control group. Meanwhile, 0.5-fold change of gene B suggests that the expression of this gene was two times less in the cyclophane group compared to its expression in the untreated group (*Romoser et al., 2011*).

## RESULTS

### Changes in gene expression

HEK cells were used to determine gene expression changes after treatment with $Fe_2PO$ and $Fe_2PC$ using an exposure concentration of 120 μM and two time points (8 or 24 h). Genes assayed in this study can contribute to four physiological response categories (inflammation, apoptosis, oxidative stress, and non-inflammatory immune responses) as shown in Table 1. Many of the genes have functions in more than one of these interdigitating pathways. According to this classification, specific results are discussed below, in the order of decreasing observed toxicological responses.

The results for the 29 genes with complete data for all samples are presented in a heat map (Table 2). Fold up-regulation or down-regulation of mRNA levels as quantified by RT-qPCR are shown compared to that of the untreated cells, and those were normalized to the housekeeping gene *B2M*.

A graphical display of unique expression is observed in scatter plots where genes whose changes in expression exceeds thresholds of 2-fold induction or 0.5-fold reduction are shown individually for 8 and 24 h (Fig. 2).

### Inflammatory response

Some of the genes included in the array employed here are intermediaries of inflammation, an immune response of vascular tissue to harmful stimuli, such as pathogens, irritants, or even damaged cells. The expression level of some of these genes changed after iron-cyclophane exposure. Among these is the ubiquitous nuclear transcription factor κB (NFκB), a pleiotropic regulator of many genes involved in immune and inflammatory processes (*Conner & Grisham, 1996*).

Most of the inflammatory changes in gene expression observed occur at the 24 h time point with either iron-cyclophane. $Fe_2PC$ induced changes in the gene expression of members of the IL-1 family, *CCR* and *TLR4* (Table 2). Importantly, pro-inflammatory *IL1A* was up-regulated early in the response at 8 h (2.20-fold) for $Fe_2PO$, and down-regulated later, after 24 h exposure. $Fe_2PC$ exposure also evoked this expression pattern. Otherwise, pro-inflammatory *IL1B* was down-regulated early in the response at 8 h for both complexes, but after 24 h exposure for $Fe_2PC$, up-regulated expression was observed (2.25 fold).

**Table 1 Classification of the genes tested by RT-qPCR array.**

| Gene symbol | Oxidative stress | Apoptosis | Inflammation | Non-inflammatory immune response |
|---|---|---|---|---|
| ADORA2A | | ✓ | ✓ | |
| CCR3 | | | ✓ | |
| CD14 | | | ✓ | ✓ |
| CHUK | | | ✓ | |
| CXCR4 | | | | ✓ |
| HMOX1 | ✓ | | | ✓ |
| IKBKB | ✓ | | ✓ | ✓ |
| IL1A | | | ✓ | ✓ |
| IL1B | | | ✓ | ✓ |
| IL36RN | | | ✓ | |
| IL37 | | | ✓ | |
| IL36G | | | ✓ | |
| IL1R2 | | | ✓ | |
| IRAK1 | | | ✓ | |
| IRAK2 | | | ✓ | |
| IRF1 | | | ✓ | |
| MIF | | | ✓ | ✓ |
| NFκB1 | | ✓ | ✓ | ✓ |
| NFκB2 | ✓ | | ✓ | ✓ |
| NLRC4 | | ✓ | | |
| PGLYRP1 | | | | ✓ |
| PGLYRP3 | | | | ✓ |
| SFTPD | | | | ✓ |
| TGFB1 | | ✓ | | |
| TLR1 | | | ✓ | ✓ |
| TLR4 | | | ✓ | |
| TNF | | ✓ | ✓ | |
| TNFRSF1A | | ✓ | ✓ | |
| TOLLIP | | | ✓ | |

## Non-inflammatory immune response

This study assesses genes not only involved with inflammatory immune responses, but also genes related to other innate and adaptive immune responses. Modification of expression levels of several genes by Fe(III) complexes indicates that pattern recognition receptors can activate antigen-presentation and therefore stimulate T cell-mediated immune responses. These effects go beyond NFκB signaling and inflammation. Pattern recognition receptors belong to the innate response, such as the peptidoglycan recognition proteins (PGLYRP) family, which were induced by Fe$_2$PC (Table 2). *CD14*, along with Toll-like receptor protein (*TOLLI*) were down-regulated at 24 h for both metal complexes. TOLLIP interacts with TLR4 during recognition of the Gram-negative bacterial lipopolysaccharide (LPS). *CD14* suppression could suggest an unexpected response of this

**Table 2 Changes in pathway-specific genes expressed in the human embryonic kidney after exposure to Fe$_2$PO and Fe$_2$PC.** Fold-change and up-regulation values greater than 2 are indicated with a ↑; fold-change values less than 0.5 and downregulation values less than −2 are indicated with a ↓. The $p$ values were calculated based on a Student's $t$-test at the $\alpha$ = 0.05 level of significance.

| Gene symbol | Fe$_2$PO complex | | T-test | Fe$_2$PC complex | | T-test |
|---|---|---|---|---|---|---|
| | 8 h | 24 h | P value | 8 h | 24 h | P value |
| ADORA2A | 0.82 | 0.33 ↓ | 0.04 | 1.41 | 0.36 ↓ | 0.15 |
| CCR3 | 0.83 | 0.75 | 0.53 | 0.78 | 3.26 ↑ | $1.2 \times 10^{-3}$ |
| CD14 | 0.83 | 0.35 ↓ | 0.01 | 2.34 ↑ | 0.52 | 0.05 |
| CHUK | 1.25 | 3.63 ↑ | $4.0 \times 10^{-3}$ | 0.71 | 4.20 ↑ | $4.5 \times 10^{-4}$ |
| CXCR4 | 1.42 | 1.04 | $8.2 \times 10^{-3}$ | 2.20 ↑ | 1.53 | $4.7 \times 10^{-3}$ |
| HMOX1 | 1.95 | 0.19 | $1.4 \times 10^{-5}$ | 1.82 | 0.15 ↓ | 0.01 |
| IKBKB | 1.95 | 0.42 ↓ | $2.1 \times 10^{-4}$ | 1.92 | 0.37 ↓ | 0.02 |
| IL1A | 2.20 ↑ | 1.19 | 0.04 | 1.01 | 0.19 ↓ | 0.03 |
| IL1B | 0.82 | 1.21 | 0.04 | 0.78 | 2.25 ↑ | $4.3 \times 10^{-3}$ |
| IL36RN | 0.82 | 1.24 | 0.03 | 0.79 | 3.88 ↑ | 0.01 |
| IL37 | 0.85 | 3.16 ↑ | $6.9 \times 10^{-3}$ | 0.89 | 6.60 ↑ | 0.01 |
| IL36G | 0.83 | 0.47 ↓ | 0.03 | 0.79 | 0.41 ↓ | 0.12 |
| IL1R2 | 1.09 | 2.08 ↑ | 0.02 | 1.60 | 2.49 | $7.1 \times 10^{-3}$ |
| IRAK1 | 1.84 | 0.28 ↓ | $7.2 \times 10^{-5}$ | 4.17 ↑ | 0.28 ↓ | $4.0 \times 10^{-3}$ |
| IRAK2 | 1.61 | 0.50 ↓ | $2.3 \times 10^{-3}$ | 3.38 ↑ | 0.49 ↓ | 0.02 |
| IRF1 | 0.74 | 0.42 ↓ | 0.02 | 1.38 | 0.90 | 0.28 |
| MIF | 1.06 | 0.28 ↓ | $1.1 \times 10^{-3}$ | 1.18 | 0.28 ↓ | 0.05 |
| NFκB1 | 0.98 | 2.66 ↑ | 0.01 | 0.64 | 2.46 ↑ | 0.04 |
| NFκB2 | 0.86 | 0.46 ↓ | 0.02 | 1.21 | 0.41 ↓ | 0.08 |
| NLRC4 | 2.49 ↑ | 5.58 ↑ | 0.01 | 1.87 | 6.22 ↑ | 0.01 |
| PGLYRP1 | 0.50 | 1.12 | 0.01 | 1.09 | 4.07 ↑ | 0.04 |
| PGLYRP3 | 0.83 | 1.57 | 0.02 | 1.54 | 4.09 ↑ | $2.4 \times 10^{-3}$ |
| SFTPD | 0.68 | 1.72 | 0.01 | 1.05 | 2.74 ↑ | $9.4 \times 10^{-3}$ |
| TGFB1 | 0.99 | 0.65 | 0.07 | 2.31 ↑ | 0.73 | 0.01 |
| TLR1 | 0.82 | 2.48 ↑ | $9.6 \times 10^{-3}$ | 0.79 | 1.55 | 0.03 |
| TLR4 | 0.82 | 1.33 | 0.03 | 0.79 | 1.63 | 0.56 |
| TNF | 0.82 | 1.57 | 0.02 | 0.73 | 0.52 | 0.16 |
| TNFRSF1A | 0.97 | 0.62 | 0.06 | 0.97 | 0.60 | 0.23 |
| TOLLIP | 1.34 | 0.20 ↓ | $8.4 \times 10^{-6}$ | 1.52 | 0.17 ↓ | 0.03 |

receptor for the perception of moieties other than conserved LPS patterns in bacterial cell walls (*Gilroy et al., 2004*).

## Apoptotic response

Genes related to the programed cell death pathway were interrogated to determine their responses to metal cyclophane treatment. The apoptotic gene for the adenosine receptor *ADORA2A* was strongly down-regulated by both Fe$_2$PO and Fe$_2$PC. Tumor necrosis factor TNF binds its receptor ligand to start the external apoptotic pathway, and

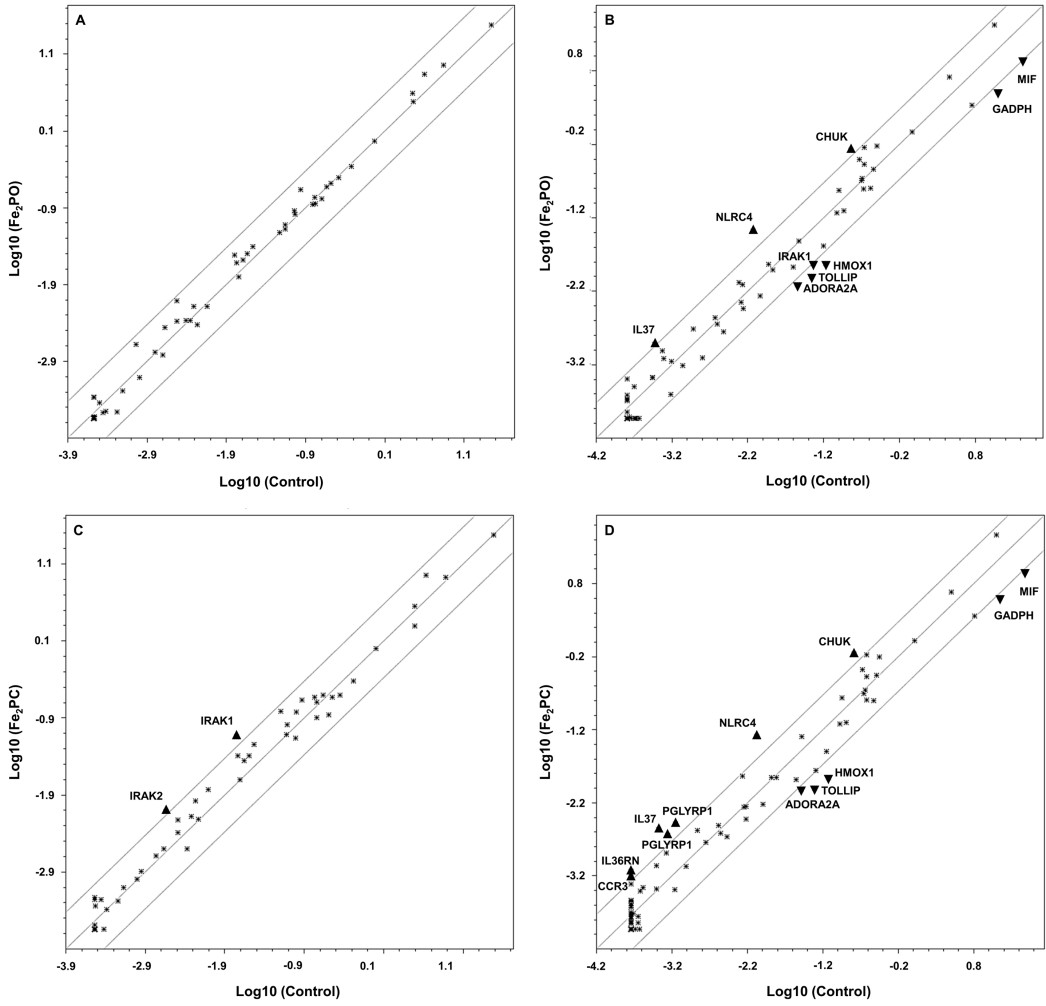

**Figure 2 Scatterplots of modulated genes in human embryonic kidney exposed to Fe₂PO and Fe₂PC.**
(A) 120 µM Fe$_2$PO exposed cells vs. unexposed cells at 8 h; (B) 120 µM Fe$_2$PO exposed cells vs. unexposed cells at 24 h; (C) 120 µM Fe$_2$PC exposed cells vs. unexposed cells at 8 h; (D) 120 µM Fe$_2$PC exposed cells vs. unexposed cells at 24 h. ▲ Indicate genes that were up-regulated at least 2.0-fold compared to the control; ▼ represent genes downregulated at least 0.5-fold compared to the control.

down-regulation of the TNF receptor superfamily member 1A (*TNFRSF1A*) was also observed for both metal complexes after the 24 h treatment.

Nevertheless, *NLRC4* was strongly up-regulated, suggesting the possibility of apoptosis after 24 h treatment. The caspase recruitment domain (CARD) containing protein 4 (NLRC4) interacts with caspase 1 and NOS2 (*Damiano, Newman & Reed, 2004*; *Damiano, Oliveira & Welsh, 2004*), and was strongly induced at 8 and 24 h by Fe$_2$PO and 24 h by Fe$_2$PC (Table 2). NFκB is essential for the transcriptional upregulation of these apoptotic pathways, and even though cytosolic levels of *NFKB1* mRNA were upregulated after 24 h exposure, transcript levels of *NFKB2* were down at both time points. It has been demonstrated in some conditions that NFκB acts as an anti-apoptotic protein,

suggesting an unknown mechanism by which NFκB protects against cell death (*Ghosh, May & Kopp, 1998*).

## Stress responses by reactive oxygen intermediates and/or metals

Since iron is complexed into the host molecules, we looked into stress response genes for changes in gene expression. The results show that neither iron-cyclophane complex led, to changes in expression of genes related to cellular stress such as *HMOX1*, *IKBK* and *NFKB*d. For all of these genes, down-regulated effects were observed for both $Fe_2PO$ and $Fe_2PC$ metal complexes, growing to a strong effect at the 24 h exposure time point.

## DISCUSSION

Several studies have described the activation of inflammatory responses by metal particles that induce the release of cytokines such as IL-1 and IL-18 (*Love et al., 2012*; *Romoser et al., 2012*). The activation of the inflammatory immune responses by metal complexes or nanoparticles is an appropriate toxicological response. Many mechanisms of recognition and signaling of danger have been conserved against fundamental chemical properties and insult, both biological and toxicological (*Criscitiello et al., 2013*). In this work, the expression of several genes demonstrated the capability of bioactive $Fe_2PO$ and $Fe_2PC$ metal complexes to induce either an innate or adaptive immune response (*Criscitiello & De Figueiredo, 2013*).

Changes in the relative expression of several genes in this study warranted special consideration. The expression of *NLRC4* was up-regulated both early and late by $Fe_2PO$ and $Fe_2PC$. *NLRC4*, a NOD-like cytosolic receptor, is expressed in innate immune cells. It senses bacterial flagellin and indirectly, the type III secretion system, triggering a response by assembling an inflammasome complex that leads to caspase-1 activation and pyroptosis (*Qu et al., 2012*). However, it is also involved in post-translational processing/ secretion of IL-1β and IL-8 via the activation of caspase-1 (*Carvalho et al., 2012*). Therefore, the activation of *NLRC4* by $Fe_2PO$ and $Fe_2PC$ results in the up-regulation of the genes encoding NFκB1, the proinflammatory cytokine IL-1, and IL1R2 at 24 h. However, this up-regulation of proinflammatory signals also may result from the activation of *IRAK1* and *IRAK2* observed at 8 h. IRAK1 and IRAK2 are protein kinases that act as downstream mediators of TLR signaling and induce the activation of TRAF6 and MAP kinases to promote NFκB translocation to the nucleus where it can promote the up-regulation of several cytokines (*Zhu & Mohan, 2010*).

The increased levels of mRNA expression of *PGLYRP1* and *PGLYRP3* at 24 h could be associated with the pro-inflammatory environment and NFκB induced by $Fe_2PO$ and $Fe_2PC$ (*Lee et al., 2012*; *Uehara et al., 2005*). Also, the levels of TLR1 and TLR4 mRNA expression increased at 24 h of culture, and this may also be the result of the levels of NFκB. Even though the levels of NLRC4 were substantially increased and this gene is an apoptotic inductor, the other apoptotic-associated gene robustly quantified (*ADORA2A*) was down-regulated. This protector effect could be associated with the production of anti-inflammatory cytokines, IL37 was induced by $Fe_2PO$ and $Fe_2PC$ at 24 h and the gene

encoding TGF-β was expressed at 8 h after $Fe_2PC$ stimulation. IL-37, a member of the IL-1 cytokine family, has been identified as a natural suppressor of inflammatory and other innate immune responses (*Boraschi et al., 2011*; *Nold et al., 2010*). It is highly expressed in tissues with inflammatory processes, and it and protects from the damage caused by chronic inflammation, and it can also ameliorate autoimmune diseases (*Ji et al., 2014*; *McNamee et al., 2011*).

The possibility of identifying a specific pathway with these data is complicated, as the genes we focused upon in this study have pleiotropic functions along several intertwining conduits. Nevertheless, the toxicological effects in which the modulated genes are involved (oxidative stress, apoptosis, inflammatory response and/or non-inflammatory immune response) after $Fe_2PO$ and $Fe_2PC$ treatment let us relate the physiology observed in the cells to the NFκB pathway. Corroborating the perturbation of this key transcription factor pathway, two of the most active genes affected were *CHUK* and *IL37*, both implicated in the modulation of NFκB.

CHUK, an inhibitor of NF-kB kinase subunit alpha (IKK-α), along with IKK-β, phosphorylates the IκB protein marking it for degradation via ubiquitination and allowing NFκB to translocate to the nucleus. Once activated, NFκB regulates a plethora of genes that are implicated in many critical cellular processes, as has been rigorously described (*Romoser, Criscitiello & Sayes, 2014*). This fundamental relating of transcription factors to changes in gene expression, lead us to a deeper understanding of how extracellular agents modifies cellular behavior and survival (*O'Neill & Kaltschmidt, 1997*).

## CONCLUSIONS

Inflammatory responses were elicited by the exposures to the two cyclophane iron complexes tested. After 8 h of exposure, the cells showed higher relative expression of several inflammatory mediators such as those genes encoding interleukin 1, IRAK molecules that transduce signals between interleukin 1 receptors and the NFκB pathway, and the LPS pattern recognition receptor CD14. After 24 h of exposure, regulatory genes begin to counter the inflammation, as some genes involved in oxidative stress, apoptosis and non-inflammatory immune responses also come into play. Both $Fe_2PO$ and $Fe_2PC$ induced similar immunogenetic changes in transcription profiles, but equal molar doses of $Fe_2PC$ resulted in more robust responses.

The harmlessness of $Fe_2PO$ and $Fe_2PC$ was been observed previously in human and murine systems (*Salazar-Medina et al., 2013*; *Salazar-Medina et al., 2017*) and the current work's findings improve our understanding of how cyclophane metal complexes can exert bioactive properties as antioxidants, without inducing cell mortality.

More work is required to discern the roles of the metal and cyclophane structures in these responses, and also to confirm the observed transcriptional effects at the level of protein expression and eventually response in the exposed animal. Additional studies querying dose-response, querying more time points and confirming expression of the translated product are now mandated to confirm the findings of this study and inform drug development decisions.

### Funding

This work was supported by the Mexico National Research Council for Science and Technology (CONACYT) grants Cátedras Patrimoniales, Red Química Supramolecular 294810 and basic science grant CB-2014-236216. Support was also received by the Texas A&M-CONACYT Collaborative Grant 2011–050 Program. Universidad de Sonora funded a sabbatical leave at Departamento de Investigación en Polímeros y Materiales to Dr. Islas-Osuna. The funders had no role in study design, data collection and analysis, decision to publish, or preparation of the manuscript.

### Grant Disclosures

The following grant information was disclosed by the authors:
Mexico National Research Council for Science and Technology (CONACYT): 294810.
Basic Science: CB-2014-236216.
Texas A&M-CONACYT Collaborative: 2011–050.
Universidad de Sonora.

### Competing Interests

Rogerio R. Sotelo-Mundo is an Academic Editor for PeerJ. Christie M. Sayes is employed by RTI International. The authors declare that they have no competing interests.

### Author Contributions

- Alex J. Salazar-Medina conceived and designed the experiments, performed the experiments, analyzed the data, prepared figures and/or tables, authored or reviewed drafts of the paper, and approved the final draft.
- Enrique F. Velazquez-Contreras conceived and designed the experiments, authored or reviewed drafts of the paper, and approved the final draft.
- Rocio Sugich-Miranda conceived and designed the experiments, performed the experiments, authored or reviewed drafts of the paper, and approved the final draft.
- Hisila Santacruz conceived and designed the experiments, authored or reviewed drafts of the paper, and approved the final draft.
- Rosa E. Navarro conceived and designed the experiments, authored or reviewed drafts of the paper, and approved the final draft.
- Fernando Rocha-Alonzo conceived and designed the experiments, analyzed the data, authored or reviewed drafts of the paper, and approved the final draft.
- Maria A. Islas-Osuna conceived and designed the experiments, analyzed the data, authored or reviewed drafts of the paper, and approved the final draft.
- Patricia L. Chen conceived and designed the experiments, performed the experiments, authored or reviewed drafts of the paper, and approved the final draft.
- Sarah G.B. Christian conceived and designed the experiments, performed the experiments, authored or reviewed drafts of the paper, and approved the final draft.

- Amelia A. Romoser conceived and designed the experiments, performed the experiments, authored or reviewed drafts of the paper, and approved the final draft.
- Scott V. Dindot conceived and designed the experiments, analyzed the data, authored or reviewed drafts of the paper, and approved the final draft.
- Christie M. Sayes conceived and designed the experiments, performed the experiments, authored or reviewed drafts of the paper, and approved the final draft.
- Rogerio R. Sotelo-Mundo conceived and designed the experiments, analyzed the data, prepared figures and/or tables, authored or reviewed drafts of the paper, and approved the final draft.
- Michael F. Criscitiello conceived and designed the experiments, performed the experiments, analyzed the data, prepared figures and/or tables, authored or reviewed drafts of the paper, and approved the final draft.

## Data Availability

The qPCR data is available in the Supplemental Files.

## Supplemental Information

Supplemental information for this article can be found online at http://dx.doi.org/10.7717/peerj.8956#supplemental-information.

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
