# Peer review of "Immune response of human cultured cells towards macrocyclic Fe2PO and Fe2PC bioactive cyclophane complexes"

_PeerJ, doi:10.7717/peerj.8956_

## Round 0.1 · original submission · Major Revisions

The paper is very interesting although there are a number of issues that should be amended as sugegsted by the referees. Furthermore I would also add a comparison between the activity of the Fe(III) complexed cyclophanes and the 'free' compounds.

Reviewer 1 ·

Basic reporting

Information available in the literature on the immune response to exposure of cyclophanes is very scarce. Therefore the work presented in this paper, dealing with the immune response of cultured human cells (Human embryonic kidney (HEK) 293T cells) when exposed to Fe(III) complexes of two different cyclophanes short-named by the authors Fe2PO and Fe2PC is a valuable contribution to a little-explored field.
There are several aspects in the paper that should be improved: The English, especially in the abstract and the introduction is rather poor.
I would like to disagree with the use/abuse of the term ‘biomimetic’. Here I find it to be totally inappropriate as common sense identifies as biomimetic molecules that have significant structural similarity to biomolecules (e.g. a peptide made up of non-naturally occurring amino acids). If one adopts the criteria mentioned in this paper to define a molecule as biomimetic, then even sodium chloride is ‘biomimetic’. Hence we have an adjective that does not mean anything because it can be applied to everything. I think the authors do not need to use fashionable (but in this case meaningless) adjectives to ‘sell’ their work.
I think it is unacceptable that one can go through the whole paper without seeing the structural formulas of the investigated compound (although a reference (Salazar-Medina AJ., 2013) showing these is cited.

Experimental design

I noticed that the dosage used for the tests is quite high (120 mM) but there is no indication as to why such dosage was selected. There is no dose/response investigation. I think the authors should explain why, and also mention why they selected (HEK) 293T cells for their tests.

Validity of the findings

There is an articulated discussion on the correlations between certain genes being over or down-regulated and the related biological pathways. The data appear consistent. However, one is left to wonder as to what could be seen if the cyclophanes ‘as such’ and not their Fe(III) complexes were used in the experiments. The authors recognize this weak point in one of their conclusive remarks. Therefore, why was nothing done about it? An explanation should be provided.

Additional comments

These are covered in the above fields

Reviewer 2 ·

Basic reporting

no comment

Experimental design

More experimental methods should be employed, such as WB or Elisa.

Validity of the findings

no comment

Additional comments

Major revision before publish
Major comments:
1. The mechanism study about the thermal effect on cells is not accepted. Authors just tested two proteins and speculate the potential mechanism. It is strongly suggested to conduct more cell culture to block the signal pathway and test the downstream gene expression of the stem cells related to osteogenic differentiation.

2. There is no quantitative and statistics analysis for the results. For example, authors claimed that “The gene for the adenosine receptor (ADORA2A) is related to apoptosis, and it was strongly down-regulated by…”. The statistics analysis should be performed and P value should be provided.

3. Author tried to evaluate the immune gene expression changes in human cells exposed to metal particles. However, why is Fe2PO or Fe2PC? Authors should give more information about the reason they used Fe2PO and Fe2PC in the introduction.

4. It is an uncompleted work. It can’t be get these conclusions just depend on the gene expression. Western blot or Elisa should be performed to evaluate the expression of relevant proteins.

Minor comments:
1. Authors should carefully organize the manuscript. Many inconsistent descriptions were existed. For instance, “eight” should be corrected as 8 In line 174.
2. There is a writing error in line 48

---

## Round 0.2 · Major Revisions

Dear Authors,
Although the chemical part of the work has been properly amended and accepted by referee N.1, some biological conclusions should be drawn in order to publish the paper within PeerJ. I suggest to ask some colleagues for a Western blot or ELISa test as required by Referee nr. 2.

Reviewer 2 ·

Basic reporting

The structure of the article conformed to an acceptable format now.

Experimental design

More experimental methods should be employed, such as WB or Elisa.

Validity of the findings

The data on which the conclusions are still absence.

Additional comments

The authors adressed most of the problems, but some critical ones still yet to be done.
1.As I said, it can not be got the conclusions made in this study just based on genes expression. Authors refused to add experiments and deleted the relevant conclusion. However, this kind of correction make the paper lack of central topic and important conclusions.
2."Conclusion" should be re-eidted and exhibited the most important findings in this study. Currently, much unrelevant contents were involved in Conclusion part.

---

## Round 0.3 · accepted · Accept

I understand the authors' rebuttal letter. I think the article can be published within PeerJ